# Design and Experiment of a Harvesting Header for Wide–Narrow-Row Corn

**Wenxue Dong** [†], **Yingsi Wu** [†], **Fei Liu** *[ID], **Hengtong Hu, Jianguo Yan, Hongbin Bai** [ID] **and Xuan Zhao**

College of Mechanical and Electrical Engineering, Inner Mongolia Agricultural University, Hohhot 010010, China; dong18403045597@163.com (W.D.); yingsi888@imau.edu.cn (Y.W.); kyl2521697844@163.com (H.H.); yanyjg@126.com (J.Y.); bhb81571@163.com (H.B.); 15849316897@163.com (X.Z.)

* Correspondence: afei2208@imau.edu.cn
† These authors contributed equally to this work.

**Abstract:** Aiming to solve the problems of the large harvesting loss and low harvesting efficiency of wide- and narrow-row corn harvesting header in China, a method for the side installation of a header is proposed. A wide–narrow-row corn harvesting header with high working efficiency and low harvesting loss was designed. The collision energy balance equation of corn ears was established. The analysis shows that the kinetic energy change before and after the collision between the ear and the picking plate is the main cause of the shedding of and damage to corn kernels. Based on this, the main structural parameters of the corn harvesting header were designed. Based on the principle of Box–Behnken test, the response surface test was designed. The effects of the plant feeding speed, feeding angle, and rotation speed of a stem pulling roller on harvesting performance were analyzed. The best combination of working parameters was determined: The plant feeding speed was 1.08 m/s, the feeding angle was 52.46°, and the rotation speed of the stem pulling roller was 835.25 r/min. At this time, the grain damage rate was 1.09% and the grain loss rate was 0.14%. The corresponding parameters are verified by experiments. The results show that the grain damage rate was 1.12% and the grain loss rate was 0.14%. The optimization results are essentially consistent with the verification results, which meet the requirements of corn harvesting performance.

**Keywords:** corn; header; wide–narrow-row planting; low loss and high efficiency

## 1. Introduction

Corn is a multi-purpose crop that can be used for food, feed, processing, and energy [1–5]. The proportion of the sown area and yield of corn among China's grain crops increased from 17.1% and 19.5% (1980) to 36.4% and 40.3% (2022) [6]. Ensuring high corn yield is of great significance for maintaining national food security [7]. In the semi-humid and semi-arid regions of northern China, wide–narrow-row planting patterns are mostly used for maize. The existing large-scale harvesting header cannot adapt to the wide–narrow-row planting mode with different row spacing [8]. However, the existing small, hand-held, single-row corn harvester header and double-row corn harvester header have problems, such as low harvest efficiency and large grain loss [9,10]. Therefore, the development of a low-loss and high-efficiency corn harvesting header is the key to solving the above problems.

In the harvest process, the collision impact between corn ears and the ear picking device is the main reason for the falling off and breaking of the header grains [11–14]. Research shows that the combined picking device of a stem-pulling roller and picking plate causes less damage to ear grains [15]. However, the stem-pulling roller speed is higher than that of the stem-pulling roller in the roller-type picking device [16], resulting in a more severe collision between corn ears and the ear picking device, making it easier for the grains to fall off. In order to reduce the impact of corn ear collision during ear picking, Drago GT designed an automatic adjustment device for a picking plate, which

can adaptively adjust the gap between ear picking plates. It can effectively reduce both the impact force of corn ears and the loss of maize shedding and crushing. The Oxbo 50 series corn header adopts the combination of a conical roller and bending snapping plate to reduce the pulling speed of the ear during the ear picking process, thereby lowering the impact force of the corn ear while ensuring efficiency [17]. In addition, it is possible to reduce the possibility of grain shedding and crushing loss by using an elastic buffer device and flexible material to alleviate the impact between the corn ear and the picking plate [18–21]. This changes the structural parameters of the stem roller and the working parameters of the header. However, the above method is only suitable for large harvesters. Although the single-row or double-row small harvesters used in the wide–narrow-row planting area of corn can reduce loss using the above methods, it cannot meet the efficiency requirements of the harvest.

In order to achieve a low-loss and high-efficiency harvest in a wide- and narrow-row planting area of maize, the energy balance equation of a corn ear was established, and the causes of grain shedding and breakage during ear picking were analyzed. The side installation method of header was proposed, the main structural parameters of the header were designed, and the working parameters were optimized. The conclusion of this study can provide a theoretical reference for reducing the loss of corn harvest headers and improving harvest efficiency.

## 2. Materials and Methods

### 2.1. Structure and Principle of Operation

In order to meet the requirements of high-efficiency and low-loss harvesting in wide–narrow-row maize planting areas, the side installation method of a corn header is proposed, as shown in Figure 1a. This method can effectively increase the length, feeding amount, and feeding angle of the header, thereby improving harvest efficiency and reducing harvest loss. The main structure of a wide–narrow-row corn harvester header includes a feeding chain, stem-pulling roller, picking plate, and a divider and shield, as shown in Figure 1b.

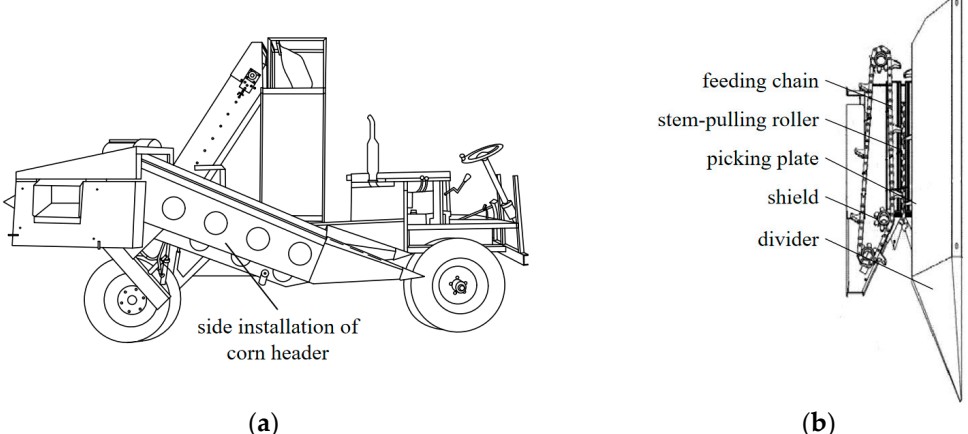

(**a**)             (**b**)

**Figure 1.** Corn harvester and header diagram: (**a**) Corn harvester diagram; (**b**) header diagram.

The corn plants to be harvested are separated from the other corn plants by a divider during the operation. The feeding chain at the end of the grain divider actively leads the corn plant to the ear picking device. The feeding corn stalk is pulled down by the stem-pulling roller. The smaller gap of the picking plate only allows the corn stalk to pass through. The corn ear is separated from the corn stalk under the obstruction of the picking plate. The picked ear is transported to the ear peeling and conveying device under the action of the feeding chain.

### 2.2. Stress Analysis of Ear

This design adopts the combined snapping device of stem-pulling roller and picking plate. The separation of ear and stem was achieved by blocking the ear using the picking plate. It can be seen that the collision between the ear and the picking plate is the main source of force on the ear. The force analysis of the ear during the ear picking process is shown in Figure 2.

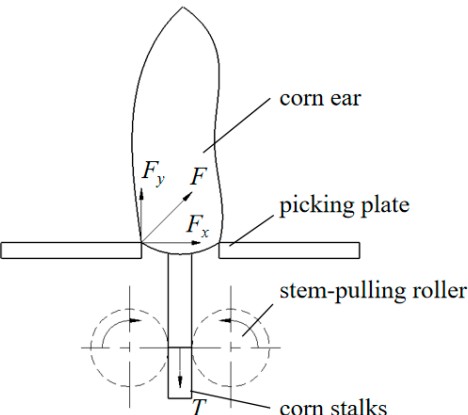

**Figure 2.** Analysis of ear force in the process of the picking ear.

When picking the ear, the stem-pulling roller produces a downward external force T on the corn stalk, i.e., the force of the stem on the ear. Under the action of the blocking force F of the picking plate on the ear and the pulling force T of the corn stalks on the ear, the ear is separated from the stalks to complete the ear picking. It can be seen that the main force of the corn ear is the contact between the bottom of the ear and the picking plate. Grain shedding and crushing are mainly concentrated at the bottom of the ear, as shown in Figure 3.

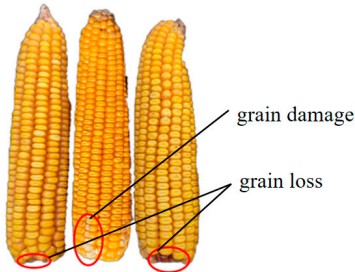

**Figure 3.** Corn ear damage map.

### 2.3. Establishment of Ear Energy Balance Equation

According to the principle of energy balance, the corn plant and the picking mechanism are regarded as a material system sealed by a closed bread. For the contact collision model between the ear and the picking mechanism, there are the following relationships:

$$E_i = E_{o1} + E_{o2} + E_{o3} \tag{1}$$

In the formula, $E_i$ is the input energy, J; $E_{o1}$ is the energy consumed by an irreversible process, J; $E_{o2}$ is other consumed energy, J; and $E_{o3}$ is the kinetic energy of the system, J.

It is assumed that the collision is at the same height before and after, and the picking plate is in a static state relative to the ear. The input energy $E_i$ is the kinetic energy before the collision of the ear. $E_{o1}$ is the energy consumed by grain shedding and crushing. $E_{o3}$ is the kinetic energy of the ear after contact collision. Therefore, the ear energy balance equation is as follows:

$$E_i - E_{o3} = E_{o1} + E_{o2} \tag{2}$$

It can be seen that the change in ear kinetic energy before and after the collision is transformed into the energy consumed by grain damage, grain shedding, and other consumed energy (thermal energy, acoustic energy, etc.). Among these, the kinetic energy before collision is as follows:

$$E_i = \frac{1}{2}mv_0^2 \tag{3}$$

Then, $E_{o3}$ is expressed as:

$$E_{o3} = \frac{1}{2}mv_0^2(\sin^2 \alpha + e^2 \cos^2 \alpha) \tag{4}$$

This is embedded as follows:

$$E_{o1} = K(E_i - E_{o3}) \tag{5}$$

In the formula, $K$ is the proportional coefficient of dimension one. The energy consumed by grain loss and damage is as follows:

$$E_{o1} = \frac{1}{2}Kmv_0^2(1 - e^2)\cos^2 \alpha \tag{6}$$

In the formula, $m$ is the quality of the ear, kg; $v_0$ is the movement speed of the ear before the collision, m/s; $\alpha$ is the angle between the relative velocity of the ear and the common normal line before the collision (i.e., the feeding angle of the corn plant), °; $e$ is the collision recovery coefficient between the ear and the contact part.

It can be seen that the change in kinetic energy before and after the collision between the ear and the picking plate is the main reason for the shedding and breaking of maize grains. Therefore, the ear energy balance equation is established. It is helpful to explore the influence of the snapping mechanism on grain shedding and crushing and provide a theoretical reference for the design of corn harvesting header.

*2.4. Pull Stem Roll Design*

2.4.1. Stem-Pulling Roller Diameter

The size of stem-pulling roller directly affects the efficiency of stalk grabbing. In order to achieve effective grasping of corn stalks, it is necessary to ensure that the stem-pulling roller does not have too large grasping clearance to grab the stalk.

In order to ensure that the stem-pulling roller grabs the corn stalk without grabbing the corn ear, its diameter should meet the following requirements:

$$\frac{d_j - h}{1 - \frac{1}{\sqrt{1 + \mu_j^2}}} \leq D \leq \frac{d_g - h}{1 - \frac{1}{\sqrt{1 + \mu_g^2}}} \tag{7}$$

In the formula, $D$ is the diameter of stem-pulling roller, mm; $d_j$ is the diameter of corn stalk, mm; $d_g$ is the diameter of the large end of the ear, mm; $h$ is the gap between two stem-pulling rollers, mm; $\mu_j$ is the grabbing coefficient of the stem-pulling roller to the corn stalk; $\mu_j$ is the grabbing coefficient of the stem-pulling roller to the corn ear. $\mu_j = \mu_g = 0.7 \sim 1.1$.

With the relevant data, we can obtain the following:

$$(d_j - h) \leq D \leq (3 \sim 3.5)(d_g - h) \tag{8}$$

This is calculated as 89 mm $\leq D \leq$ 98 mm

Combined with the actual situation, it should be ensured that the stem-pulling roller can better grasp the stem without grasping the ear. In order to achieve the best drawing efficiency and the spatial arrangement of the header platform, the diameter of the stem-pulling roller is determined to be 90 mm.

### 2.4.2. Length of the Stem-Pulling Roller

The length design of the stem-pulling roller needs to meet two conditions, adapting to different earing heights and ensuring that multiple corn plants are picked at the same time. The working diagram for this is shown in Figure 4.

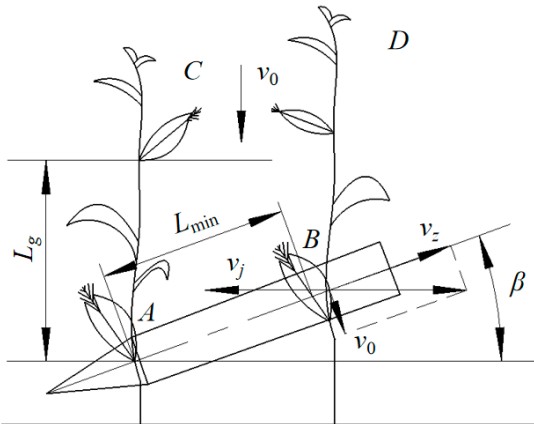

**Figure 4.** Stem-pulling roller work diagram: A, B, C and D are corn ears; $L_{\min}$ is the shortest working length of stem-pulling roller, mm; and $v_z$ is the relative speed of corn ear and stem-pulling roller, m/s.

In order to ensure that the highest and lowest corn ears and multiple corn plants are picked at the same time, the feeding amount of the header should be increased, while the working length L of the stem-pulling roller should be satisfied:

$$L \geq L_g \sin \beta \tag{9}$$

$$L \geq \frac{n_y d}{\cos \beta} \tag{10}$$

In the formula, $L_g$ is the height difference between the highest ear and the lowest ear, mm; $\beta$ is the horizontal inclination angle of the stem roller, °; $n_y$ is the number of corn plants; and $d$ is corn plant spacing, mm.

As shown in Figure 5, the length comparison diagram of the stem-pulling roller is shown. According to Formula (6), the greater the feeding angle of the corn plant, the lower the energy consumed by grain shedding and crushing. When the feeding angle increases from 30° to 60°, the energy consumed by grain shedding and crushing decreases from 2.3 J to 0.8 J. From Figure 6, it can be seen that when the fixed point and the stem-pulling roller are at a certain height from the ground, increasing the length of the lifting roller can increase the feeding angle. This can reduce the energy consumption of grain shedding and damage and decrease the losses in the header.

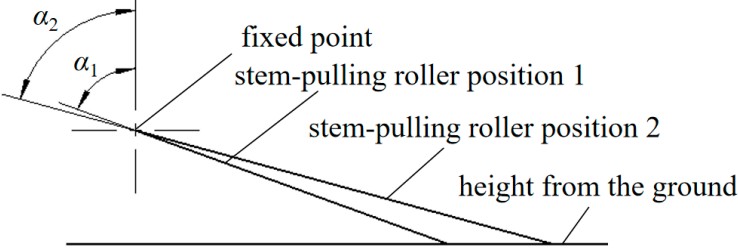

**Figure 5.** Comparison of stem-pulling roller length.

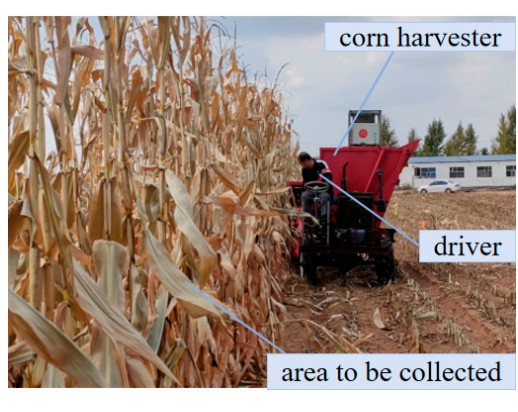
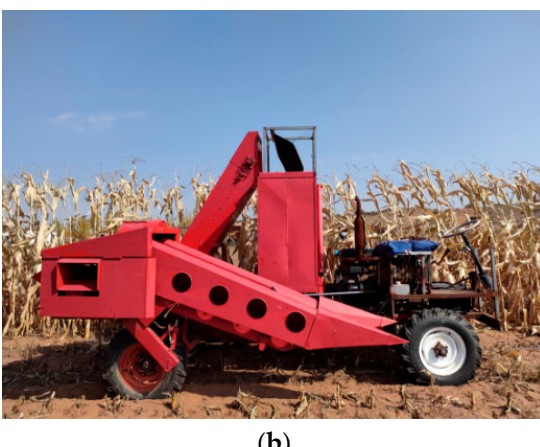

(**a**)                                                                   (**b**)

**Figure 6.** Field trial process: (**a**) test process diagram; (**b**) wide–narrow-row corn harvester.

In order to improve the picking performance of lodged plants and reduce collision losses, the length of the stem-pulling roller for this machine was ultimately determined to be 1200 mm.

### 2.4.3. Stem-Pulling Roller Rotational Speed

The rotational speed of the stem-pulling roller has a significant impact on the grabbing effect of corn stalks. When the rotational speed of the stem-pulling roller is too low, there is a tendency for relative slippage between the corn stalks and the stem-pulling roller, resulting in a poor grabbing effect. Conversely, when the rotational speed of the stem-pulling roller is too high, there is a greater force between the corn ears and the ear picking plate, leading to increased ear losses. Based on the establishment of the parallelogram rule for the forward speed of the machine and the rotational speed of the stem-pulling roller, as shown in Figure 5, the rotational speed of the stem-pulling roller is determined to be the following:

$$n = \frac{v_0}{\pi D} \tag{11}$$

$$v_0 = \frac{v_j}{C \sin \beta} \tag{12}$$

$$n = \frac{v_j}{C \pi D \sin \beta} \tag{13}$$

In the formula, $n$ is the rotational speed of stem-pulling roller, r/min; $C$ is the proportional coefficient, 0.7; $v_0$ is the linear speed of stem-pulling roller, m/s; $v_j$ is the forward speed of the machine, m/s.

Among these, when the forward speed of the machine is 6 km/h, the diameter of the stem-pulling roller is 90 mm, and the working angle of the header is 25~45°. The rotation speed range of the stem-pulling roller is 722~1203 r/min.

### 2.5. Test Conditions and Methods

The trial was scheduled for October 2023. The experiment was conducted in the experimental field of Hanlong Landscaping Technology Co., Ltd. in Zhasak Town, Ejin Horo Banner, Ordos City, Inner Mongolia Autonomous Region, China. The corn yield in the experimental field is relatively high, approximately 15,000 kg/hm$^2$. The experimental field is located in a dry area in the northwest. The soil in the planting area is sandy soil with a water content of about 11%. It was sunny and the temperature was about 10 °C at harvest. The planting pattern is wide–narrow-row planting, with a row spacing of 700 mm for the wide rows and 400 mm for the narrow rows. The length of the field is about 100,000 mm. The experimental process is shown in Figure 6.

The maize variety used in the experiment was Huamei 1. A good number of 1000 seeds were weighed on an electronic balance. The corn kernels were placed in an intelligent moisture tester to measure the moisture content, and the average value was repeated three times. The physical characteristics of corn in the experimental field are as follows: The grain moisture content is about 33.3%. The 1000-grain weight is about 355.3 g. The moisture content of the stalk is about 58%. The lowest ear height is 890 mm. The plant lodging rate is about 5%. The ear drooping rate is about 70%.

According to the GBT21961-2008 [22] "corn harvester-test method", GBT21962-2008 [23] "corn harvester-technical conditions" and NY/T645-2002 [24] "corn harvester quality evaluation technical specifications", the corn harvester was tested. The main equipment used during the testing process includes a 0–5 m tape measure, a TL-4 intelligent moisture tester, a DM6236P tachometer (Victor Archives, Camden, NJ, USA), and an SNJ-10002 electronic balance (SNJ Precision Automotive, Warwick, Australia).

The total quality values of corn kernels, damaged—(loss, obvious cracks, and broken skins) and lost (falling inside the machine and on the ground)—were obtained in each test. The grain damage rate and grain loss rate were calculated using Formulas (13) and (14), respectively.

$$Z_s = \frac{W_s}{W_i} \times 100\% \tag{14}$$

$$Z_l = \frac{W_l}{W_i} \times 100\% \tag{15}$$

In the formula, $Z_s$ is the grain breakage rate, %; $Z_l$ is grain loss rate, %; $W_s$ is the quality of damaged grains, g; $W_l$ is the loss of grain quality, g; and $W_i$ is the total grain mass, g.

### 2.5.1. Single-Factor Test

To verify the feasibility of the lateral installation method of the header and the accuracy of the energy balance equation for corn ears, a single-factor experiment was conducted. When the feeding speed of the maize plant was 1.2 m/s and the rotation speed of stem-pulling roller was 850 r/min, the feeding angles of maize plant were set to 35, 45, 55, 65 and 75°, respectively. When the feeding speed of corn plant was 1.2 m/s and the feeding angle of corn plant was 55°, the rotation speeds of the stem-pulling roller were set as 550, 700, 850, 1000, 1150 r/min, respectively. Finally, a scientific drawing and data analysis software Origin 2018 developed by OriginLab company (Wellesely/Newton, MA, USA) was used to draw the harvest performance curve.

### 2.5.2. Response Surface Test

In order to optimize various parameters of the corn harvester and improve the quality of harvesting, a three-factor and three-level response surface experiment was conducted using the factors of plant feeding speed, feeding angle, and stem-pulling roller speed, with the grain loss rate and grain breakage rate as the experimental indicators. The experimental factor coding table is shown in Table 1.

**Table 1.** Test factors and levels.

| Level | Test Factors | | |
| --- | --- | --- | --- |
| | Plant Feeding Speed A (m/s) | Feeding Angle B (°) | Stem-Pulling Roller Speed C (r/min) |
| 1 | 0.6 | 45 | 700 |
| 0 | 1.2 | 55 | 850 |
| −1 | 1.8 | 65 | 1000 |

A multi-objective optimization analysis was carried out using Design-Expert 10 software with larger pass rate and smaller leakage rate as the objective functions. The best

combination of operating parameters derived from the optimization was used as test parameters for field validation tests.

## 3. Results

### 3.1. Single Factor Test Results

The impact of plant feeding angle on harvesting performance is shown in the Figure 7a. As the feeding angle increases, the grain breakage rate gradually decreases. When the feeding angle is small, the collision energy between the ear and the ear picking plate is greater, leading to more grain detachment. As the feeding angle increases, the grain loss rate decreases. However, when the feeding angle exceeds 55°, the increased feeding angle raises the header height, causing partially lodged plants to be incompletely picked, rapidly increasing the grain loss rate.

The impact of stem-pulling roller speed on harvesting performance is shown in Figure 7b. As the stem-pulling roller speed increases, the downward pulling speed of the corn ears increases, resulting in a higher collision energy between the ears and the ear picking plate, leading to increased grain detachment and subsequently increasing the grain loss rate. At lower stem-pulling roller speeds, the contact time between the ears and the ear picking mechanism is longer, resulting in a higher grain breakage rate. However, as the stem-pulling roller speed increases, the grain breakage rate gradually decreases. When the stem-pulling roller speed exceeds 750 r/min, the increased speed raises the downward pulling speed of the ears, leading to a continuous increase in collision energy with the ear picking plate, causing an increase in the grain breakage rate.

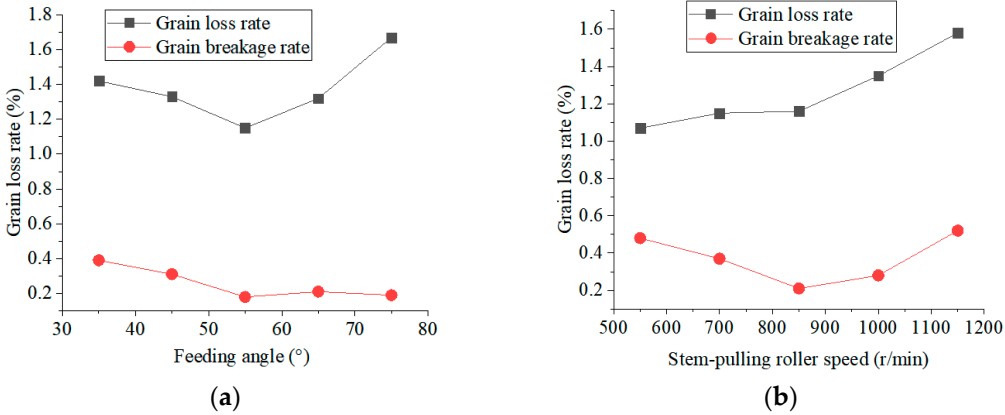

**Figure 7.** Harvest performance curve: (**a**) Harvest performance curves at different feeding angles; (**b**) Harvest performance curves at different stem-pulling roller speeds.

Single-factor experiments have shown that increasing the feeding angle appropriately and reducing the stem-pulling roller speed can improve harvesting performance. At the same time, they have validated the accuracy of the ear energy balance model.

### 3.2. Response Surface Test Results

The results of response surface test are shown in Table 2.

**Table 2.** Results of response surface test.

| Test Number | Test Factors | | | Test Indicators | |
|---|---|---|---|---|---|
| | **A** | **B** | **C** | **Grain Loss Rate** $Y_1$ **(%)** | **Grain Breakage Rate** $Y_2$ **(%)** |
| 1 | 0 | 1 | 1 | 1.57 | 0.53 |
| 2 | −1 | 0 | 1 | 1.38 | 0.51 |
| 3 | 1 | −1 | 0 | 1.47 | 0.39 |
| 4 | 0 | 0 | 0 | 1.09 | 0.11 |

**Table 2.** *Cont.*

| Test Number | Test Factors | | | Test Indicators | |
|---|---|---|---|---|---|
| | **A** | **B** | **C** | **Grain Loss Rate $Y_1$ (%)** | **Grain Breakage Rate $Y_2$ (%)** |
| 5 | 1 | 1 | 0 | 1.56 | 0.48 |
| 6 | 0 | 0 | 0 | 1.06 | 0.15 |
| 7 | −1 | −1 | 0 | 1.28 | 0.28 |
| 8 | 0 | −1 | 1 | 1.38 | 0.26 |
| 9 | 1 | 0 | −1 | 1.48 | 0.52 |
| 10 | 1 | 0 | 1 | 1.62 | 0.42 |
| 11 | −1 | 1 | 0 | 1.43 | 0.35 |
| 12 | 0 | 0 | 0 | 1.11 | 0.17 |
| 13 | 0 | 1 | −1 | 1.42 | 0.31 |
| 14 | 0 | −1 | −1 | 1.35 | 0.32 |
| 15 | −1 | 0 | −1 | 1.45 | 0.24 |
| 16 | 0 | 0 | 0 | 1.12 | 0.18 |
| 17 | 0 | 0 | 0 | 1.15 | 0.16 |

### 3.2.1. Analysis of Variance

The results of response surface analysis of variance are shown in Table 3. The regression analysis of the test results was performed by Design-Expert 10, a software for performing experimental design (DOE) provided by Stat-Ease Inc. (Minneapolis, MN, USA) The regression equation of grain loss rate $Y_1$ and grain damage rate $Y_2$ is as follows:

$$Y_1 = 1.11 + 0.0738A + 0.0575B + 0.0263C - 0.015AB + 0.0525AC + 0.04BC + 0.1858A^2 + 0.1433B^2 + 0.1907C^2 \quad (16)$$

$$Y_2 = 0.154 + 0.0537A + 0.0587B + 0.035C + 0.005AB - 0.0925AC - 0.0575BC + 0.138A^2 + 0.083B^2 + 0.1305C^2 \quad (17)$$

**Table 3.** Variance analysis of response surface test. * indicates significant.

| Source of Variance | Grain Loss Rate $Y_1$ (%) | | | | | |
|---|---|---|---|---|---|---|
| | Sum of Squares | Freedom | Mean Square | F-Value | *p*-Value | Significance |
| Model | 0.5230 | 9 | 0.0578 | 63.3 | <0.0001 | ** |
| A | 0.0435 | 1 | 0.0435 | 47.63 | 0.0002 | ** |
| B | 0.0264 | 1 | 0.0313 | 34.21 | 0.0006 | ** |
| C | 0.0055 | 1 | 0.0078 | 8.55 | 0.0222 | * |
| AB | 0.0009 | 1 | 0.0009 | 0.9851 | 0.354 | |
| AC | 0.0110 | 1 | 0.011 | 12.07 | 0.0104 | * |
| BC | 0.0064 | 1 | 0.0036 | 3.94 | 0.0875 | |
| $A_2$ | 0.1453 | 1 | 0.1532 | 167.7 | <0.0001 | ** |
| $B_2$ | 0.0864 | 1 | 0.0805 | 88.09 | <0.0001 | ** |
| $C_2$ | 0.1532 | 1 | 0.1453 | 159.02 | <0.0001 | ** |
| Residuals | 0.0055 | 7 | 0.0009 | | | |
| Fail to fit | 0.0010 | 3 | 0.0006 | 0.5531 | 0.6729 | |
| Error | 0.0045 | 4 | 0.0011 | | | |
| Total | 0.5285 | 16 | | | | |
| **Source of Variance** | **Grain Breakage Rate $Y_2$ (%)** | | | | | |
| | Sum of Squares | Freedom | Mean Square | F-Value | *p*-Value | Significance |
| Model | 0.3093 | 9 | 0.0344 | 53.52 | <0.0001 | ** |
| A | 0.0231 | 1 | 0.0231 | 35.99 | 0.0005 | ** |
| B | 0.022 | 1 | 0.022 | 34.34 | 0.0006 | ** |

**Table 3.** *Cont.*

| Source of Variance | Grain Breakage Rate $Y_2$ (%) | | | | | |
|---|---|---|---|---|---|---|
| | Sum of Squares | Freedom | Mean Square | F-Value | *p*-Value | Significance |
| C | 0.0136 | 1 | 0.0136 | 21.2 | 0.0025 | ** |
| AB | 0.0001 | 1 | 0.0001 | 0.1557 | 0.7049 | |
| AC | 0.0342 | 1 | 0.0342 | 53.3 | 0.0002 | ** |
| BC | 0.0196 | 1 | 0.0196 | 30.52 | 0.0009 | ** |
| $A_2$ | 0.0876 | 1 | 0.0876 | 136.44 | <0.0001 | ** |
| $B_2$ | 0.0248 | 1 | 0.0248 | 38.62 | 0.0004 | ** |
| $C_2$ | 0.065 | 1 | 0.065 | 101.23 | <0.0001 | ** |
| Residuals | 0.0045 | 7 | 0.0006 | | | |
| Fail to fit | 0.0016 | 3 | 0.0005 | 0.7192 | 0.5905 | |
| Error | 0.0029 | 4 | 0.0007 | | | |
| Total | 0.3138 | 16 | | | | |

Note: * indicates significant, $0.05 < p < 0.1$; ** indicates highly significant, $p < 0.01$.

The analysis of variance results indicates that the feeding speed, feeding angle, and stem-pulling roller speed all have significant effects on the two indicators, and there are interactions between some of these factors. The feeding speed and the stem-pulling roller speed had an interactive effect on the grain loss rate and the grain damage rate. The feeding angle and stem-pulling roller speed have an interactive effect on the grain damage rate. The regression models for the two evaluation indicators have a highly significant goodness of fit, with misfit terms of $p > 0.05$, indicating the absence of other factors influencing the evaluation indicators.

3.2.2. The Influence of Various Factors on Each Test Index

The primary and secondary order of influence was determined by calculating the contribution rate of factors. The mathematical model is as follows:

$$y = a_0 + \sum_{i=1}^{m} a_i x_i + \sum_{i \leq j, i=1}^{m} a_{ij} x_i y_j \tag{18}$$

The contribution rate was calculated according to the F value corresponding to each item of the objective function. Its expression is as follows:

$$\Delta_j = \delta_j + \frac{1}{2} \sum_{\substack{i \to 1 \\ i \neq j}}^{m} \delta_{ij} + \delta_{jj} \tag{19}$$

In the formula, when $F \leq 1$, $\delta = 0$, and when $F \geq 1$, $\delta = 1 - \frac{1}{F}$; $\Delta_j$ is the contribution rate of factors to the test index; $\delta_j$ is the first-order contribution rate of the jth factor; $\delta_{jj}$ is the contribution rate of the second term of the jth factor; $\delta_{ij}$ is the contribution rate of the jth factor interaction term.

The factor contribution rate of each index is shown in Table 4. The order of contribution rate to grain loss rate and grain damage rate was feeding speed, feeding angle, and stem-pulling roller speed. The greater the contribution rate, the greater the influence of experimental factors on the index.

**Table 4.** Contribution of each factor to the experimental indicators.

| Test Indicators | Contribution Rate of Each Factor | | |
| --- | --- | --- | --- |
| | **A** | **B** | **C** |
| $Y_1$ | 2.93 | 2.83 | 2.82 |
| $Y_2$ | 2.95 | 2.92 | 2.91 |

### 3.2.3. The Influence Law of Each Factor on Each Test Index

Based on the regression equation, the surface plots depicting the impact of factors with significant pairwise interactions on evaluation indicators were drawn, as shown in Figure 8.

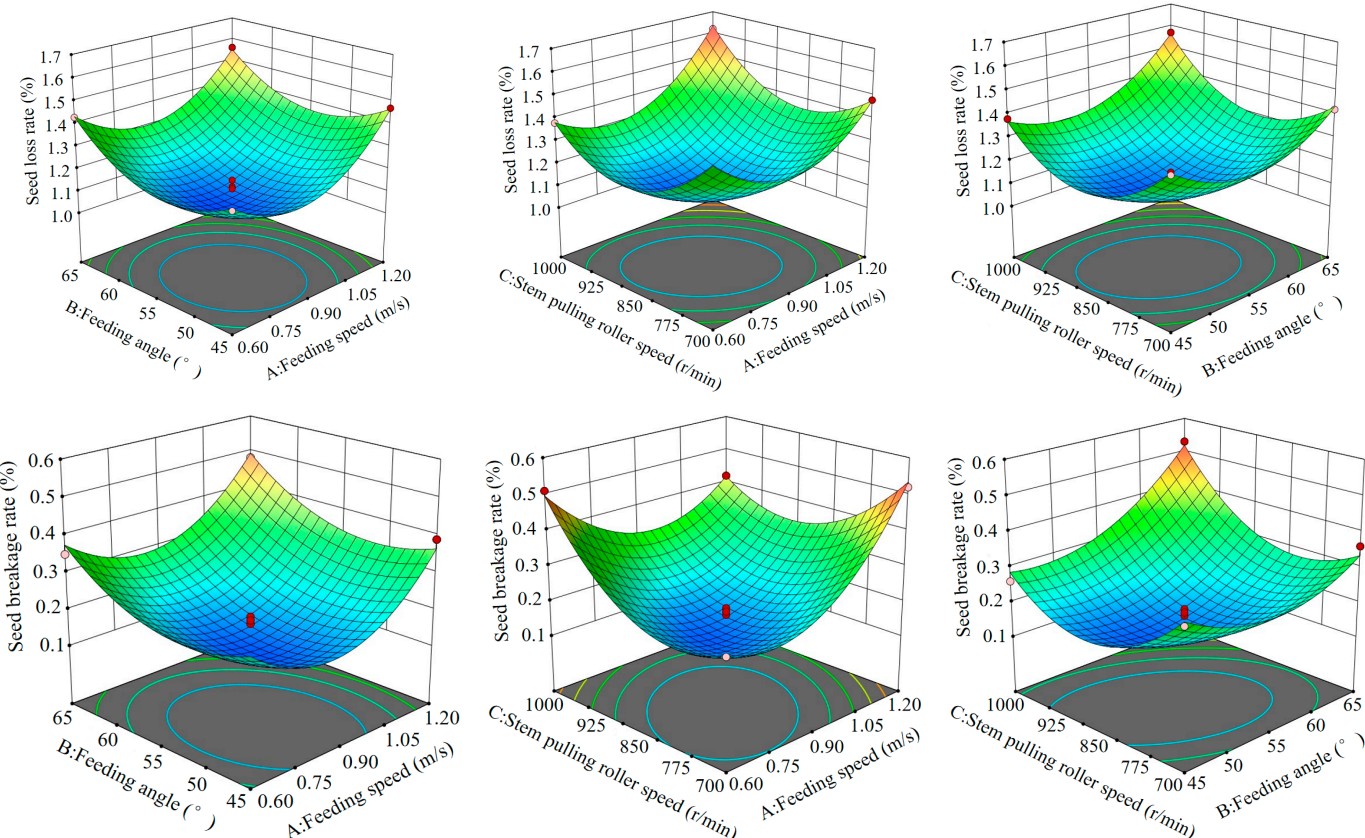

**Figure 8.** Response surface diagram of factor interaction.

When the stem-pulling roller speed is 850 r/min and the plant feeding angle is certain, both the grain loss rate and grain breakage rate first decrease and then increase with the increase in the feeding speed of the corn plant. When the feeding speed is constant, the grain loss rate and grain breakage rate first decrease and then increase with the increase in the plant feeding angle. When the plant feeding angle is 55° and the stem-pulling roller speed is constant, both the grain loss rate and grain breakage rate first decrease and then increase with the increase in the corn plant feeding speed. When the corn plant feeding speed is constant, both the grain loss rate and grain breakage rate first decrease and then increase with the increase in the stem-pulling roller speed. When the corn plant feeding speed is 1.2 m/s and the stem-pulling roller speed is constant, both the grain loss rate and grain breakage rate first decrease and then increase with the increase in the plant feeding angle. When the plant feeding angle is constant, both the grain loss rate and grain breakage rate first decrease and then increase with the increase in the stem-pulling roller speed.

When the feeding speed is relatively low, the corn ears directly collide with the ear picking plate, and the impact force generated by the collision causes the grains to detach

and break. With the increase in feeding speed, the number of corn plants in the header increases, and some plant stems and leaves act as a buffer between the corn ears and the ear picking plate, reducing the collision impact between the corn ears and the ear picking plate. When the feeding speed exceeds 1.0 m/s, the feeding speed and amount increase. This causes easy congestion in the header, which will increase the probability of collision between the ears and increase grain detachment and breakage.

### 3.2.4. Parameter Optimization and Validation Results

It can be seen from the above content that the interaction between various factors has a great influence on the working performance of the header. Therefore, the reasonable matching of working parameters between factors is key to reducing harvest loss. Using Design-Expert software, a multi-objective optimization analysis will be conducted with the goal of achieving a higher qualification rate and a lower leak rate. The objective function and constraints are as follows:

$$\begin{cases} \min Y_1(A, B, C) \\ \min Y_2(A, B, C) \\ \text{s.t.} \begin{cases} 0.6 \, \text{m/s} \leq A \leq 1.8 \, \text{m/s} \\ 45° \leq B \leq 65° \\ 700 \, \text{r/min} \leq C \leq 1000 \, \text{r/min} \end{cases} \end{cases} \tag{20}$$

The optimal working parameters obtained through calculations are as follows: the plant feeding speed is 1.08 m/s, the feeding angle is 52.46°, and the stem-pulling roller speed is 835.25 r/min. Under these parameters, the grain breakage rate is 1.09% and the grain loss rate is 0.14%.

Field validation tests were conducted using the following test parameters: the plant feeding speed is 1.08 m/s, the feeding angle is 52.46°, and the stem-pulling roller speed is 835.25 r/min. Field trial results validate this optimization, showing a grain breakage rate of 1.12% and a grain loss rate of 0.14%, which is consistent with the optimized results, meeting the requirements for corn harvesting performance.

## 4. Discussion

Studying the loss and damage mechanisms of corn harvesting platforms is crucial for the design and optimization of corn harvesting platforms. Geng Aijun et al. [25] established a mathematical model for the force on the ears of corn and found that the frictional force and acceleration on the ears during picking significantly affect grain detachment and damage. Cheng Chengying et al. [26], through comparative experiments, discovered that the structural parameters of the picking roller and the picking ridge are the main factors affecting grain detachment and damage. Chen Meizhou et al. [27] used high-speed photography and found that the "retention" of the ears and bouncing on the picking roller are the main causes of secondary damage to the ears. However, the above studies did not consider the impact of energy changes on ear damage. Therefore, this paper established an energy balance equation for the ears and found that the change in kinetic energy before and after the collision between the ear and the picking plate is the main cause of inducing corn grain detachment and damage.

This research shows that lengthening the header can increase the feeding angle of the plants while maintaining the header height above the ground, reducing the energy consumption of grain detachment and damage, thereby reducing harvest losses. Additionally, extending the header can increase its capacity, allowing for higher feeding speeds, while maintaining the stem-pulling roller speed at a relatively low level, reducing harvest losses and increasing harvesting efficiency. Existing loss reduction measures, such as adding cushioning devices or using flexible materials to reduce the force on the ears [28–30], improving the shape of the picking plate [31,32] and the structural parameters of the stem-pulling roller [33–37], and implementing stem-pulling roller gap adjustment, can reduce harvest losses to some extent. These measures are highly effective when applied to large corn

harvesters, but when used in single- or double-row harvesters in narrow-row corn planting areas, they can significantly reduce harvesting efficiency. The side-mounted header proposed in this paper can effectively address this issue.

## 5. Conclusions

Aiming to solve the issues of the high harvest losses and low harvesting efficiency of the wide–narrow-row corn harvesting platforms in China, a side-mounted header method was proposed, and a high-efficiency, low-harvest-loss, wide–narrow-row corn harvesting platform was designed.

(1) An energy balance model was established for the corn ears, which clarified that the change in kinetic energy before and after the collision between the ear and the picking plate is the main cause of inducing corn grain detachment and damage.

(2) Single-factor experiments showed that the side-mounted header method can effectively increase the header length, thereby increasing the feeding angle and improving the feeding speed. This can reduce harvest losses and increase harvesting efficiency. Additionally, it validates the accuracy of the energy balance model for the corn ears.

(3) Through response surface experiments, the analysis of the impact of plant feeding speed, stalk roller speed, feeding angle, and the interaction between some factors on the indicators was conducted. The optimal working parameter combination was determined to be a plant feeding speed of 1.08 m/s, a feeding angle of 52.46°, and a stalk roller speed of 835.25 r/min, resulting in a grain breakage rate of 1.09% and a grain loss rate of 0.14%. Field trials confirmed a grain breakage rate of 1.12% and a grain loss rate of 0.14%. The design complies with relevant national standards and can meet the requirements for low-loss and efficient harvesting in wide–narrow-row corn planting areas.

**Author Contributions:** Conceptualization, W.D. and Y.W.; methodology, W.D.; software, W.D.; validation, X.Z.; formal analysis, Y.W.; investigation, J.Y.; resources, F.L.; data curation, W.D. and H.H.; writing—original draft preparation, W.D.; writing—review and editing, Y.W. and H.B.; the Author of the photographs included, W.D. and H.H.; visualization, W.D. and X.Z. All authors have read and agreed to the published version of the manuscript.

**Funding:** This research was funded by the Inner Mongolia Autonomous Region Science and Technology Plan Project (2023YFHH0012) and Key R & D project of Ordos City (2022YY027).

**Institutional Review Board Statement:** Not applicable.

**Informed Consent Statement:** Not applicable.

**Data Availability Statement:** Data are contained within the article.

**Acknowledgments:** Funding acquisition, Zhao, J. supervision, Wu, L.J.; project administration, Dai. X.Q.

**Conflicts of Interest:** The authors declare no conflict of interest.

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
