# Peer review of "Design and Experiment of a Harvesting Header for Wide–Narrow-Row Corn"

_applsci, doi:10.3390/app14031309_

Round 1

Reviewer 1 Report

Comments and Suggestions for Authors

Dear authors,

the article is interesting but several modifications are necessary.

2.1, 2.2 and 2.3 have the same title . Change them accordingly (Or add information in order to be clearer)!

Line 57- without .

Line 63- are instead of is

Line 64- are instead of is

Line 68- Structure

Line 72-, thereby

Line 84- Structure

Line 113- What other energy?

Line 142- please rephrase. Maybe "With the relevant data you can obtain"

Line 162- please explain with numbers (not just the formula). I assume you have the calculations

Line 200- please add a picture of the narrow-wide corn harvester (like you added a picture with the machine)

Line 215- please explain more concisely. First you say a number of feeding angles and then just one (55)!

Line 245- insert space here so that the information is separated

Line 251- for table 2 you have to add some information. You can not simply just say these are the results. Which is the highest and the smallest value and why? What other information from this table is relevant?

Line 257- insert space here

Line 258- please explain with more information all the data in table 3. The results need to be fully explained in order to make sense and to prove your point.

Line 275- please add more explanations here. This is a part of the results. You have the data but you have to explain the results

Line 302- Move on the next page

Line 343- of the

Les than half of your references are from the last 5 years. Please add at least another 10 references from the last 5 years.

Author Response

We thank this referee for his comments. We are also very grateful for her/his many technical comments helped us significantly improve our work. Your opinion is very good, thank you very much. I have revised the manuscript based on your comments and marked them in red. Regarding Table 2, -1 indicates low level and 1 indicates high level. The specific values represented by -1 and 1 are given in Table 1. Table 2 relates to the information in the table, response surface testing designed by Design-Expert software.

Reviewer 2 Report

Comments and Suggestions for Authors

The manuscript submitted for evaluation contains quite interesting test results with the maize harvesting attachment. What I do not understand is the information about the narrow row spacing. Maize is a light-loving crop and therefore the row spacing is 70 or 75 cm. Is a different spacing used in the authors' study area. If so, why?

Materials and Methods

In the methodology, the authors write that the yield was 1000 kg. This is not yield but grain weight (yield = weight/area). It is difficult to conclude, as the authors write, that it was high because they do not state from which area it came.

I could not find information on which year the survey was carried out. When was the collection made? The authors also do not give the name of the maize variety tested and its characteristics. It would also be advisable to include a brief description of the maize production technology and the course of weather conditions during the growing season.

The weight of 1000 grains is given, not 100 grains as it is in the manuscript. How were grain moisture and weight determined (apparatus, methodology)?

The Author's name is missing under the photographs.

The manuscript does not contain information on the manufacturer of the statistical software used to compile the data.

References

The number of publications is quite small (28). Some are more than 10 years old, so they should be removed.

Author Response

We thank this referee for his comments. We are also very grateful for her/his many technical comments helped us significantly improve our work.

Reviewer 3 Report

Comments and Suggestions for Authors

The manuscript “Design and experiment of wide-narrow row corn harvesting header” is a very interesting study. Just a few recommendations to the authors:

1-      In the Abstract, Results and Conclusion sections, the authors describe that: “the best combination of working parameters: The plant feeding speed was 1.08 m/s, the feeding angle was 52.46°, and the rotation speed of the stem pulling roller was 835.25 r/min. At this time, the grain damage rate was 1.09% and the grain loss rate was 0.14%. The corresponding parameters are verified by experiments. The results showed that the grain damage rate was 1.12% and the grain loss rate was 0.14%.”; however, the methodology to determine this is described in results and not in methodology. Likewise, in the methodology it is not clear where the authors describe the procedure to corroborate these data in the field. It is suggested that the authors describe this in the methodology.

2-      The authors are recommended to improve the figures included in materials and methods and results. Specifically include information that allows the figures to explain themselves. The authors should consider improving figures 1 and 3, making figures 2 and 4 similar, since they describe different aspects but of the same process, figure 5 does not indicate how the process of the ears of the upper part of the plants, and Figure 7 is not informative about the process described.

3-      The authors are recommended to consider the information included in the Materials and Methods section, since several points correspond to results of their analyses.

4-      In section 2.5.1, lines 215 and 217, authors are recommended to specify which information they are referring to with the word "respectively" on each line.

5-      In Table 1, is the value of 11.2 correct as plant feeding speed?

6-      Authors are recommended to standardize subtitles according to the guide for authors, for example if they begin with a lowercase or uppercase letter, as seen in subtitles 2.2, 2.3 and 2.4.

7-      Authors are recommended to standardize references according to the guide for authors; differences are observed in references 7, 8, 10, 13, 15, 17, 18, 19, 21 and 28.

8-      Authors are recommended to include references 22 and 23, the information they have corresponds to the journal's model file.

Author Response

(The authors gave the same response as above.)

Round 2

Reviewer 2 Report

Comments and Suggestions for Authors

The authors have incorporated some of the suggested corrections. I still did not find information on soil conditions, the course of weather conditions during the growing season, the characteristics of the maize variety tested, the Author of the photographs included and the manufacturer of the statistical software. There are entries in References that are more than 10 years old.

Author Response

We thank this Referee for her/his comments. We also appreciate it very

much her/his technical comments, which have helped us improving quite significantly our work.

I have modified it according to your opinion and marked it red.
